TOPICAL REVIEW

# Impact of cancer cachexia on respiratory muscle function and the therapeutic potential of exercise

Ben T. Murphy [iD], John J. Mackrill and Ken D. O'Halloran [iD]

*Department of Physiology, School of Medicine, College of Medicine and Health, University College Cork, Cork, Ireland*

Handling Editors: Laura Bennet & Scott Powers

The peer review history is available in the Supporting Information section of this article (https://doi.org/10.1113/JP283569#support-information-section).

**Abstract**   Cancer cachexia is defined as a multi-factorial syndrome characterised by an ongoing loss of skeletal muscle mass and progressive functional impairment, estimated to affect 50–80% of patients and responsible for 20% of cancer deaths. Elevations in the morbidity and mortality rates of cachectic cancer patients has been linked to respiratory failure due to atrophy and dysfunction of the ventilatory muscles. Despite this, there is a distinct scarcity of research investigating the

**Ben Murphy** graduated with a BSc in Sport and Exercise Sciences from the University of Limerick in 2018 and an MSc in Exercise Physiology from Loughborough University in 2020. He is currently undertaking his PhD studies at University College Cork in collaboration with Dr John Mackrill and Professor Ken O'Halloran. He has a particular interest in the mechanisms which underlie skeletal muscle adaptation to exercise, and how understanding of these mechanisms can inform therapeutic implementation of exercise to combat disease. His studies will focus on exercise as an intervention in novel mouse models of cancer cachexia with a view to the adoption of strategies for implementation in people with cancer.

*The Journal of Physiology*

structural and functional condition of the respiratory musculature in cancer, with the majority of studies exclusively focusing on limb muscle. Treatment strategies are largely ineffective in mitigating the cachectic state. It is now widely accepted that an efficacious intervention will likely combine elements of pharmacology, nutrition and exercise. However, of these approaches, exercise has received comparatively little attention. Therefore, it is unlikely to be implemented optimally, whether in isolation or combination. In consideration of these limitations, the current review describes the mechanistic basis of cancer cachexia and subsequently explores the available respiratory- and exercise-focused literature within this context. The molecular basis of cachexia is thoroughly reviewed. The pivotal role of inflammatory mediators is described. Unravelling the mechanisms of exercise-induced support of muscle via antioxidant and anti-inflammatory effects in addition to promoting efficient energy metabolism via increased mitochondrial biogenesis, mitochondrial function and muscle glucose uptake provide avenues for interventional studies. Currently available pre-clinical mouse models including novel transgenic animals provide a platform for the development of multi-modal therapeutic strategies to protect respiratory muscles in people with cancer.

(Received 18 July 2022; accepted after revision 9 September 2022; first published online 17 October 2022)

**Corresponding author** Ken D. O'Halloran: Department of Physiology, School of Medicine, College of Medicine and Health, University College Cork, Cork, Ireland. Email: k.ohalloran@ucc.ie

**Abstract figure legend** Cancer cachexia is characterised by profound loss of skeletal muscle mass. Exercise has the potential to counteract several factors that contribute to muscle wasting. Atrophy and dysfunction of the ventilatory muscles can result in respiratory compromise and ultimately failure. There is a scarcity of research investigating structure and function of the respiratory musculature in cancer. Pre-clinical and clinical studies focused on mechanisms of exercise-induced support of muscle are required to drive the development of multi-modal therapeutic strategies to protect respiratory muscles in people with cancer.

## Introduction

Cancer cachexia has been defined as 'a multi-factorial syndrome characterised by an ongoing loss of skeletal muscle mass, with or without a loss of fat mass, that cannot be fully reversed by conventional nutritional support and leads to progressive functional impairment' (Fearon et al., 2011). Lack of consensus on an appropriate definition, classification and diagnostic criteria hindered early research (Fox et al., 2009). Therefore, Fearon et al. (2011) developed the above working definition and the internationally recognised framework illustrated in Fig. 1.

Incidence rates of the condition have been estimated at 50–80% of cancer patients, with variation observed between cancer types (Gordon et al., 2005). Sufferers of cancer cachexia experience an amalgamation of severe adverse effects. The nature of the induced weight loss causes progressive decline in muscle performance (Weber et al., 2009), exercise capacity (Jones et al., 2012) and activity level (Dodson et al., 2011; Wilcock et al., 2008), resulting in accelerated functional decline (LeBlanc et al., 2015), disability with regard to activities of daily living (Naito et al., 2017), increased fatigue (Stewart et al., 2006) and consequently poorer quality of life (Dewys

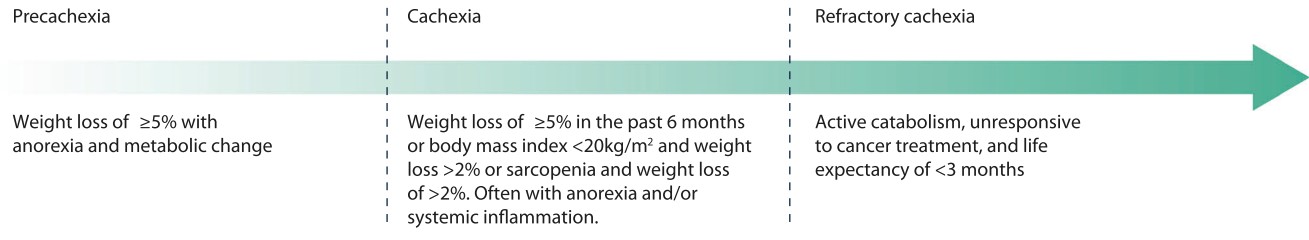

| Precachexia | Cachexia | Refractory cachexia |
|---|---|---|
| Weight loss of ≥5% with anorexia and metabolic change | Weight loss of ≥5% in the past 6 months or body mass index <20kg/m² and weight loss >2% or sarcopenia and weight loss of >2%. Often with anorexia and/or systemic inflammation. | Active catabolism, unresponsive to cancer treatment, and life expectancy of <3 months |

The Journal of **Physiology**

**Figure 1. Internationally recognised framework illustrating the continuum of cachexia progression across three clinically relevant stages**
Adapted with permission from Fearon et al. (2011).

et al., 1980). Additionally, loss of appetite and anorexia are often present (Laviano et al., 2003). Furthermore, cachexia is considered to contribute significantly toward increased morbidity and mortality in cancer populations (Tisdale, 2002), estimated to be responsible for up to 20% of cancer deaths (Argiles et al., 2014), with death reported to typically occur once weight loss exceeds 30–40% (Arthur et al., 2014). One of the leading hypotheses surrounding the link between muscle atrophy and elevated mortality considers the enhanced susceptibility of cachectic cancer patients to chemotherapy-induced toxicity (Prado et al., 2007, 2008, 2009), often leading to reductions in treatment, which subsequently impacts patient survival (Andreyev et al., 1998; Martin et al., 2013). What can be stated with conviction is that cachexia imparts a severe burden, with potentially fatal consequences, on populations of cancer patients.

In association with critical illness-related muscle weakness, respiratory muscle dysfunction in cancer has been identified as a distinct and highly relevant issue (Berger et al., 2016). Dysfunction of the inspiratory muscles, primarily the diaphragm, can result in hypoxaemia and hypercapnia. Reduced maximal inspiratory pressure in advanced cancer patients has been correlated with increased prevalence of dyspnoea (Bruera et al., 2000). Additionally, dysfunction of the expiratory muscles can inhibit airway clearance, which can increase risk of pneumonia (Gea et al., 2012; Kravitz, 2009), a major cause of respiratory failure and death in critically ill cancer patients (Azoulay et al., 2004; Windsor & Hill, 1988). Inspiratory muscle weakness is frequently observed among cancer patients (Bruera et al., 2000; Dudgeon & Lertzman, 1998; Dudgeon et al., 2001) and has been reported to contribute significantly to cachexia-associated morbidity and mortality (Feathers et al., 2003). This contribution is likely related to the observed correlation between reduced inspiratory function and a reduction in the workload necessary to induce respiratory failure (Callahan & Supinski, 2010). Unfortunately, there is a scarcity of observational reporting specific to respiratory muscle in cancer cachexia, as the majority of research in this domain has focused on limb muscle. Ultimately, despite imparting a considerable impact on patient well-being, relatively little attention has been given to the study of cachexia with the focal point of interest centred on respiratory muscle wasting and function.

A recent systematic review conducted by the American Society of Clinical Oncology aimed to produce evidence-based recommendations for the clinical management of cancer cachexia (Roeland et al., 2020). The authors concluded that insufficient evidence was available in support of pharmacological intervention. Additionally, although dietary guidance and nutritional supplementation may be of some benefit toward weight gain and improved quality of life, the depth of evidence is poor and findings are equivocal. In consideration of exercise interventions, no eligible trials were identified according to the review criteria. A lack of clinical evidence was also reported by the authors of a recent Cochrane review, who identified only four relevant studies on the effects of exercise in cachexia, hindering their ability to meaningfully interpret the limited data (Grande et al., 2021). The association between severe cancer types and the prevalence of cachexia is likely a major contributing factor to this distinct lack of clinical research. Although there is growing support for the benefits of exercise in cancer patients (Singh et al., 2020), including those with advanced disease (Lazzari et al., 2021), exercise as a therapeutic approach is yet to be recommended in incurable cancer patients (Patel et al., 2019). Additionally, despite encouraging findings, there remain considerable complications associated with the prescription of exercise in cancer populations, with primary barriers to participation reported to be treatment side effects, comorbidities and medical complications (Frikkel et al., 2020). Furthermore, the potential contraindications and complications of considerable muscle wasting cannot be dismissed. As a result, the ability for such results to inform the feasibility and safety of an exercise intervention in cancer cachexia is limited. Inclusion of metrics such as adherence and occurrence of adverse events in future studies of exercise in cancer cachexia populations would help to address such concerns. Preclinical research provides a more bountiful source of information, with murine models of cancer cachexia frequently employed, primarily the Lewis lung carcinoma model (Cai et al., 2004; Zhang et al., 2017) and the C26 colorectal adenocarcinoma model (Bonetto et al., 2016). Results from such research are promising in support of the benefits of exercise in the amelioration of the adverse effects of cancer cachexia (Bowen et al., 2015; Donatto et al., 2013; Hardee et al., 2020). However, the variety of models utilised poses a barrier to synthesis of data across individual studies, and uncertainty remains in the translatability of results from these models to humans (Talbert et al., 2019). Several conclusions can be drawn from the above information. Firstly, there is a need for clinical research examining the therapeutic capacity of exercise in cancer cachexia patients. Secondly, selection of an adequate cancer model capable of accurately representing the cachectic condition as it presents in humans is key to identification of appropriate mechanisms, and thus, therapeutic targets for clinical intervention design.

Considering the uncertainty surrounding the most appropriate therapeutic approaches for sufferers of cachexia, extensive further research toward the development of treatment strategies is needed. Past research indicates that in order to provide impactful benefit, such a strategy would likely need to be multimodal in nature, potentially combining elements of nutrition,

exercise and pharmacology. However, due to the complexity of the condition, and the continued work toward complete understanding of the mechanisms and mediating factors, careful consideration must be given to the limits of our current knowledge. Combination of ineffective strategies is unlikely to produce further benefit than said strategies alone. As such, the pursuit of an efficacious multimodal approach must be predicated on comprehensive understanding of the benefits provided by the individual elements. Additionally, the sparsity of cachexia research addressing respiratory muscle wasting and function highlights a particular need for investigations attending to this area. The current review aims to describe the mechanisms implicated in the aetiology of cancer cachexia, and subsequently collate and explore the available respiratory- and exercise-focused literature in consideration of this mechanistic context.

## Mechanisms of cachexia

Mechanistically, cachexia is a diverse condition, exhibiting a pro-inflammatory state and elevations in oxidative stress, resulting in upregulation of protein degradation, inhibition of protein synthesis and reduced insulin sensitivity (Gould et al., 2013; Tisdale, 2009). The underlying mechanisms responsible for mediating the alterations seen in these processes are numerous and far from fully elucidated. Discussion surrounding the causative roots of cachexia has addressed the condition from the sides of both atrophy and hypertrophy, assessing the stimulation and suppression of these processes, respectively. Past investigations have followed logical paths of enquiry, from describing established signalling pathways, to mediating factors of such pathways, to resultant gene expression and translation. Such explorations have greatly expanded our understanding of the mechanisms of the condition. When considered collectively, the multi-factorial inter-connecting nature of the contributing mechanisms becomes apparent. Interrelations across pathways and between components complicate the determination of the relative impact of factors which have been studied in isolation. Therefore, although evidence exists indicating the involvement of a range of factors, identifying promising avenues for future research, few conclusive statements can be made in relation to the mechanisms of cancer cachexia. Where possible, the system should be considered holistically. Isolated components which have been evaluated in terms of their influence on the cachectic state need also be considered for their impact on other mediating elements. A thorough integrative understanding of this manner is necessary in order to identify the most valid pursuits aimed at enhancing the mechanistic understanding of cancer cachexia.

**Protein degradation/skeletal muscle atrophy.** Regarding protein breakdown, the ubiquitin–proteasome system (UPS) is the primary pathway responsible for the removal of unwanted and damaged intracellular proteins (Goldberg, 2003). This is particularly the case with regard to the degradation of myofibrillar proteins in skeletal muscle (Attaix et al., 1998). The ubiquitination of proteins involves the action of the E1 ubiquitin-activating enzyme, E2 ubiquitin-conjugating enzymes and E3 ubiquitin-protein ligases, the latter of which act to closely regulate proteolysis in skeletal muscle (Kitajima et al., 2020). Of particular interest in the context of muscle degradation are two muscle-specific E3 ligases, namely muscle RING finger 1 (MuRF1) and muscle atrophy F-Box (MAFbx; also known as Atrogin-1). Interest in these genes stems from the work of Bodine, Latres et al. (2001) and Gomes et al. (2001) who detected them at low levels in resting skeletal muscle, with significant increases following immobilisation or food deprivation in mice. Considerable evidence has shown similar upregulation of MuRF1 and MAFbx mRNA across many preclinical models, and in response to every condition which induces skeletal muscle atrophy (Bodine & Baehr, 2014), including cachexia (Lecker et al., 2004). Furthermore, suppression of MuRF1 or MAFbx may support maintenance of muscle mass (Bodine, Latres et al., 2001), although this effect is not evident across all atrophy conditions (Baehr et al., 2011). This has led to further investigations into the potentially key regulatory roles of these factors. Some findings suggest MAFbx may be linked to the control of protein synthesis via degradation of MyoD and eIF3-f (Lagirand-Cantaloube et al., 2008; Tintignac et al., 2005). However, greater credence is attributed to the role of MuRF1 in the control of protein degradation via the specific targeting of myosin heavy chain and other thick filament proteins for ubiquitination (Cohen et al., 2009). Notably, evidence in clinical instances is lacking, which represents the primary limitation of such hypotheses. In addition, due to the ability of a given E3 ligase to bind to different E2 enzymes dependent on the cellular environment, it remains a possibility that the substrates of MuRF1 and MAFbx may vary as a function of the condition responsible for the induction of the muscle wasting (Bodine & Baehr, 2014). Therefore, whilst evidence indicates MuRF1 and MAFbx are implicated in the early induction and regulation of skeletal muscle atrophy, further *in vivo* research is necessary to accurately determine the mechanistic role of these factors. Upstream of these factors, expression of ubiquitin ligase is regulated by the Forkhead box O (FoxO) transcription factors (Sandri et al., 2004), whose sub-cellular location in turn is affected by Akt. Phosphorylation of FoxO transcription factors by Akt results in their translocation from the nucleus to the cytoplasm, thus dampening the transcription and subsequent production of MuRF1 and

MAFbx (Kitajima et al., 2020). Such activity illustrates the interconnected nature of the processes responsible for protein degradation and synthesis, as Akt also plays an important role in muscle hypertrophy. Under normal conditions, and when regulated appropriately, proteolysis via the UPS is critical in the prevention of disease and cellular dysfunction (Rock et al., 1994), and is necessary in the facilitation of healthy ageing. However, dysregulation and subsequent increased activity of this system is thought to contribute to the increased proteolysis evident in the cachectic condition (Fig. 2).

**Protein synthesis/skeletal muscle hypertrophy.** Several anabolic pathways facilitate the induction of muscle hypertrophy and an intracellular shift away from muscle protein degradation toward protein synthesis. The primary identified pathways are the Akt–mammalian target of rapamycin (mTOR), mitogen-activated protein kinase (MAPK) and calcium ($Ca^{2+}$)-dependent pathway (Schoenfeld, 2010). Of these, the Akt–mTOR pathway is considered to be of particular importance, with mTOR being widely known as the master regulator of skeletal muscle growth (Bodine, Stitt et al., 2001; Thomas & Hall, 1997). Thus, in addressing the contribution of alterations in protein synthesis toward the development of cachexia, suppression of this pathway is of considerable interest.

Exploring the implicated factors from the top down, insulin-like growth factor-1 (IGF-1) acts as an upstream regulator. Binding of IGF-1 to its receptor recruits insulin receptor substrate 1 (IRS1) (Bohni et al., 1999), phosphorylation of which is necessary for the activation of many downstream signalling events (Egerman & Glass, 2014), illustrating the regulatory role of IGF-1 in the modulation of protein synthesis. This

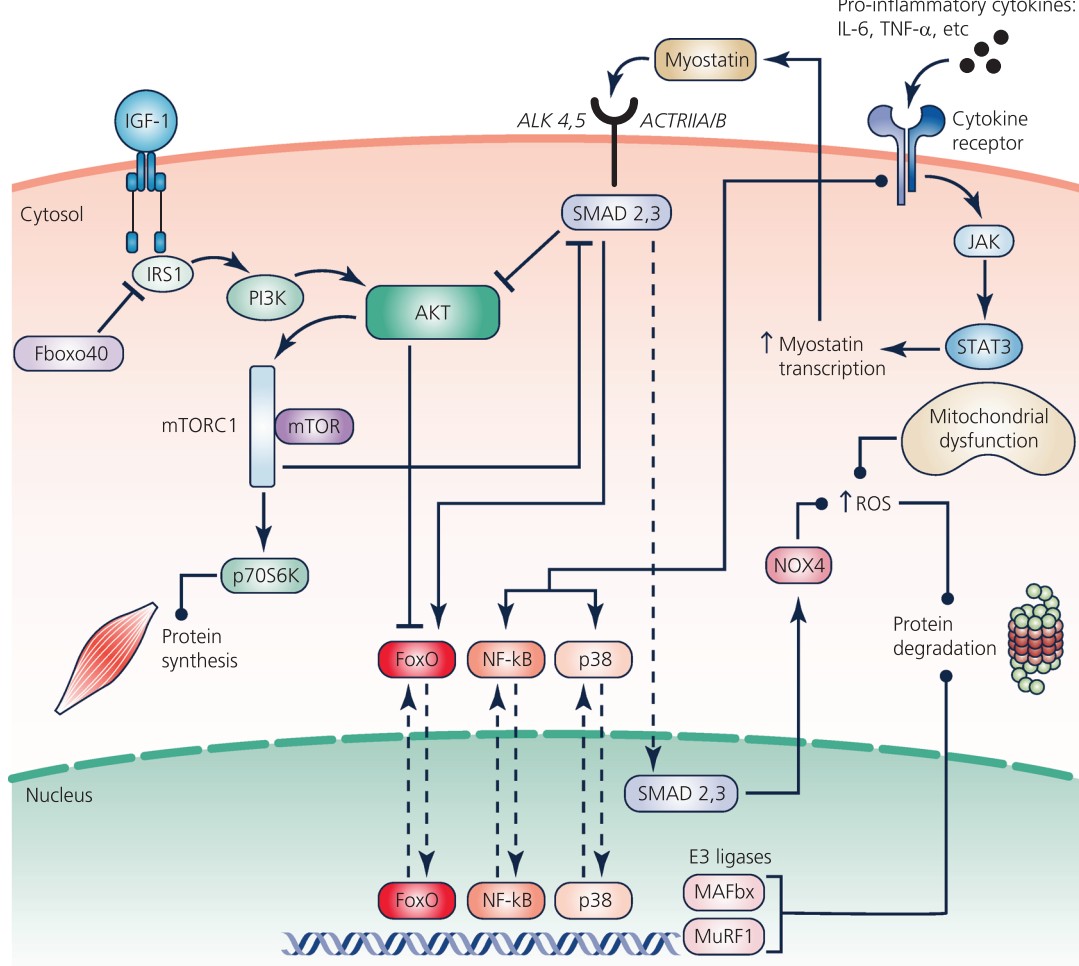

**Figure 2. Outline of the primary signalling factors and mediators implicated in the pathways of protein synthesis and protein degradation in skeletal muscle via the activation of mTOR and the ubiquitin–proteasome system, respectively**

is demonstrated in the potential for inactivation of the IGF-1 pathway via ubiquitin-mediated degradation of IRS1, which limits the downstream activation of the IRS1–phosphoinositide 3-kinase (PI3K)–Akt pathway (Shi et al., 2011). Unimpeded phosphorylation of IRS1 initiates a signalling cascade, involving recruitment and phosphorylation of PI3K, which subsequently results in Akt phosphorylation. This is of critical importance in the hypertrophy process (Rommel et al., 2001), with Akt acting both to stimulate anabolism and to inhibit catabolic signalling (Nader, 2005; Toigo & Boutellier, 2006), via FoxO phosphorylation. Delving deeper, the master regulator of muscle growth, mTOR, is a downstream target of Akt (Nave et al., 1999). Briefly, Akt, via intermediary signalling events, activates mTOR complex-1 (mTORC1), resulting in phosphorylation of $p70^{s6k}$, which promotes protein synthesis via activation of ribosomal protein S6 (Yoshida & Delafontaine, 2020). This pathway, linking IGF-1 to Akt to mTOR, has been established as the primary process of mediation in the stimulation of muscle hypertrophy (Glass, 2003).

Also of interest is the myokine myostatin, a member of the transforming growth factor-$\beta$ family which is responsible for the regulation of cell growth via hypertrophy inhibition (Lee & McPherron, 1999; McPherron et al., 1997). Myostatin acts via binding to its receptors, activin receptor IIA/activin receptor IIB and activin-like kinase-4/5. Subsequent intracellular signalling activates the transcription factors Smad2 and Smad3, initiating their translocation to the nucleus and activation of target genes (McCroskery et al., 2003). Myostatin can reduce protein synthesis by suppressing Akt or increase protein degradation by elevating FoxO transcription. There is evidence to suggest IGF-1 can maintain Akt phosphorylation when applied concurrently with myostatin to myotubes (Trendelenburg et al., 2009). This suggests that the effect of IGF-1 on mTORC1 signalling involves a feedback loop, with increased myostatin-induced Smad2 phosphorylation observed following mTORC1 inhibition. Therefore, within the complex hypertrophy signalling framework, there is apparent myostatin–Akt crosstalk, whereby IGF-1 and myostatin act as antagonists operating a level above mTOR phosphorylation, influencing the respective stimulation and repression of muscle hypertrophy. The multi-faceted influence between myostatin signalling and the Akt–mTOR pathway, in addition to its ability to elevate FoxO transcription, present an intriguing narrative for the exploration of myostatin's contribution to the development of the cachectic condition.

### The adverse influence of cachexia

*Inflammation.* Elevated systemic inflammation is a hallmark of cancer (Argiles et al., 2012; Korniluk et al.,

2017). Inflammatory cytokines are released both from the tumour and from immune cells in response to the disease (Argiles et al., 2012), and are heavily associated with the development of cachexia (Paval et al., 2022). Inflammatory cytokines mediate alterations in the UPS by promoting activation of transcription factors related to wasting (Mangner, et al., 2013; Puppa, et al., 2014), and they are the subjects of extensive research in this regard. Proinflammatory cytokines implicated in muscle wasting, such as IL-6, IL-1, IL-8, tumour necrosis factor-$\alpha$ (TNF-$\alpha$), TNF receptor-associated factor 6, tumour necrosis factor-like weak inducer of apoptosis (TWEAK), leukaemia inhibitory factor, and $\gamma$-interferon exert their influence via the nuclear factor kappa B (NF-$\kappa$B) and the p38 MAP kinase signalling pathways. Activity of these pathways is necessary for the upregulation of MuRF1 and MAFbx (Trendelenburg et al., 2009), and thus can be indicative of UPS dysregulation. Indeed, inhibition of NF-$\kappa$B activation was sufficient to prevent muscle wasting due to acute pulmonary and subsequent systemic inflammation in mice, suggesting a requirement for NF-$\kappa$B activation in the transition from inflammation to atrophy (Langen et al., 2012). Additionally, inhibition of NF-$\kappa$B significantly reduced muscle loss in tumour-bearing mice in association with downregulation of MuRF1. Initiation of muscle wasting and the subsequent development of cachexia in pre-cachectic APC$^{min/+}$ mice was linked to the overexpression of IL-6 (Baltgalvis et al., 2008). Additionally, IL-6 is suggested to contribute to the mitochondrial abnormalities associated with tumour growth, as neutralising IL-6 antibodies restored altered peroxisome proliferator-activated receptor-gamma coactivator (PGC)-1$\alpha$ levels and mitochondrial content (White et al., 2012). The cytokine TWEAK in particular has been linked to upregulation of MuRF1 and resultant breakdown of the myosin heavy chain contractile component of skeletal muscle (Mittal et al., 2010). Interestingly, inflammatory cytokines also activate the Janus kinase (JAK)–signal transducer and activator of transcription 3 (STAT3) pathway, with an induction of STAT3 phosphorylation observed in cancer (Bonetto et al., 2012). The implication of the JAK/STAT signalling pathway in the development of oxidative stress (Duan & Bai, 2020) highlights a potential interconnecting and synergistic interaction between these mediating states present in cancer cachexia.

*Oxidative stress.* Oxidative stress is produced by excessive reactive oxygen species (ROS) production, reflected by an imbalance between the manifestation of systemic ROS and ability of the system to detoxify the reactive intermediates or repair the resulting cell damage (Zorov et al., 2006). One potential mechanism responsible for this is mitochondrial dysfunction leading to uncoupling of oxidative phosphorylation, which has

been observed in Lewis lung carcinoma (LLC) mice (Tzika et al., 2013). Subsequently, mitochondrial uncoupling contributes to aberrant energy metabolism, which in turn is closely related to ROS emission (Cadenas, 2018). Mitochondrial uncoupling proteins (UCP) act to lower the electrochemical proton gradient across the mitochondrial membrane (Ricquier & Bouillaud, 2000), thus facilitating proton leak. This in turn has been shown to decrease ROS production in isolated mitochondria (Boveris & Chance, 1973). Additionally, increased ROS production was observed in the mitochondria of skeletal muscle from UCP3 knockout mice, suggesting a regulatory role of UCPs in ROS generation (Vidal-Puig et al., 2000). Notably, UCP3 is almost exclusively expressed in skeletal muscle (Sanchis et al., 1998), and evidence suggests that cancer cachexia leads to the induction of UCP2 and UCP3 via TNF-$\alpha$ (Collins et al., 2002). Indeed, significant upregulation of UCP3 was observed alongside a down-regulation of PGC-$1\beta$ and altered energy metabolism in LLC cancer cachectic mice (Constantinou et al., 2011), perhaps suggesting a regulatory antioxidative response of the system to the imposed stress of the condition. However, in parallel to decreased ROS, elevated levels of UCPs result in increases in resting energy expenditure via thermogenesis and the dissipation of energy as heat (Giordano et al., 2003). In this way, the activity of UCPs may be linked to the elevations in basal metabolic rate observed in cachexia sufferers. Additionally, increased expression of UCP3 has also been linked to the activation of the proteolytic systems in cultured myotubes (Busquets et al., 2006), suggesting a potential active role in muscle atrophy. Therefore, although there exists a theoretical argument for the targeted upregulation of UCPs with the intention of combatting conditions associated with oxidative stress, the mechanistic contribution of these factors in cachexia is still unclear.

The development of a pro-oxidative state imparts an influence on skeletal muscle hypertrophic and atrophic pathways. With regard to atrophy, mitochondrial oxidative stress enhances protein degradation via activation of the UPS and autophagy pathways (Barbieri & Sestili, 2012). In the context of hypertrophy, oxidative stress impairs the differentiation of myoblasts and myotubes, and has been implicated in muscular pathologies defined by imbalances in proliferation, such as sarcopenia and cachexia (Fulle et al., 2004). Furthermore, elevated expression of ROS has been linked to downregulation of IGF-1 signalling and induction of insulin resistance (Bashan et al., 2009). The damaging effect of elevations in free radical expression also impacts muscle function, inducing unwanted alterations in protein structure, including the sarcoplasmic reticulum $Ca^{2+}$ release channel/ryanodine receptor (Matecki et al., 2016), cross bridge kinetics or the reduction in myofilament $Ca^{2+}$ sensitivity (Powers & Jackson, 2008). In assessing

a potential pharmacological intervention, Smuder and colleagues (2020) illustrated the impact of mitochondrial dysfunction and excessive ROS production in a C26 cancer cachexia model. Tumour-bearing control animals experienced significant loss of diaphragm muscle mass and subsequent ventilatory dysfunction associated with increased ROS and inflammatory marker expression. The contributions of oxidative stress and inflammation toward muscle wasting in cancer cachexia may work in parallel, or indeed promote the action of one another, as cytokines, particularly TNF-$\alpha$, are associated with oxidative stress in skeletal muscle during cancer (Powrozek et al., 2018). Additionally, the JAK–STAT3 signalling pathway is strongly linked to IL-6 and was found to be activated in the diaphragm of a cancer cachexia mouse model, inducing oxidative stress and diaphragm atrophy and dysfunction (Bonetto et al., 2012). Activation of STAT3 also directly leads to increased expression of myostatin, caspase-3, atrogin and MuRF, further contributing to protein degradation via ubiquitination (Bonetto et al., 2012; Demaria et al., 2010; Tang et al., 2015). The myostatin-regulated transcription factor Smad3 has been implicated in the excessive production of ROS via induction of mitochondrial NADPH oxidase 4 (NOX4) (Goodman et al., 2013). Thus, increased STAT3 activation contributes to a pro-oxidative state via increases in myostatin and subsequent expression of Smad3.

## Respiratory muscle function and wasting

To collate and interpret the relevant existing studies concerning the respiratory system in cancer cachexia a systematic search of the literature was conducted. This process involved the definition of appropriate inclusion and exclusion criteria (Appendix A) from which a search strategy was developed according to the PICO tool, as recommended by the Cochrane Collaboration (Higgins et al., 2019). Indexed headings, identified via the Medical Subject Headings (MeSH) browser (https://meshb.nlm.nih.gov/search), and appropriate keywords were inputted into a table (Appendix B) and used to guide the development of a systematic search strategy. A database search using this strategy was then conducted in PubMed (Appendix C). The resultant screening of papers is illustrated in the PRISMA flow diagram (Moher et al., 2009) in Fig. D1 (Appendix D). Descriptive information including study characteristics, aims and relevant findings are presented in Table 1.

Although the extent of research investigating cachexia in the context of its impact on respiratory muscle atrophy and function is scarce, limited results have been published, and these findings support the need for further consideration in this domain. Early results were equivocal, with a number of the original studies addressing this

**Table 1. Qualitative data obtained from the studies (n = 12) isolated during a systematic search of the PubMed database**

| Author (Year) | Aim | Design | Respiratory relevant findings | |
|---|---|---|---|---|
| | | | Primary comparison | Alternative comparisons |
| Bachmann et al. (2009) | Investigate the influence of cachexia on fat, muscle and lung function in patients with pancreatic cancer | Non-cachectic PDA patients; Cachectic PDA patients | CC vs. non-CC: ↓ relative VC; ↔ absolute VC; ↔ absolute FEV1; ↔ relative FEV1 | |
| Chacon-Cabrera et al. (2014) | Assess the effects of treatment with NF-κB, MAPK or proteasome inhibitors on respiratory and limb muscle in cancer cachexia | Control; LP07 LC; LP07 LC + proteasome inhibitor; LP07 LC + NF-κB inhibitor; LP07 LC + MAPK inhibitor | CC vs. control — Diaphragm: ↓ muscle mass; ↑ protein degradation; ↓ myostatin; ↑ myogenin; ↓ MyHC | CC + proteasome vs. CC: ↔ muscle mass; ↓ protein degradation; ↓ myostatin; ↔ myogenin; ↔ MyHC. CC + NF-κB vs. CC: ↑ muscle mass; ↓ protein degradation; ↓ myostatin; ↑ myogenin; ↑ MyHC. CC + MAPK vs. CC: ↑ muscle mass; ↓ protein degradation; ↓ myostatin; ↑ myogenin; ↑ MyHC |
| Choi et al. (2013) | Assess the validity of the Lewis lung carcinoma model and examine its effects on skeletal muscle | Control; LLC | CC vs. control — Diaphragm: ↓ specific force | |
| Fermoselle et al. (2013) | Explore whether cancer cachexia alters MRC complexes and oxygen uptake in respiratory and limb muscles | Control; LP07 LC; LP07 LC + NAC; LP07 LC + NF-κB inhibitor; LP07 LC + MAPK inhibitor | CC vs. control — Diaphragm: ↓ muscle mass; ↔ citrate synthase; ↓ MRC complex I, II, IV; ↓ oxygen consumption | CC + NAC vs. CC: ↔ muscle mass; ↔ citrate synthase; ↔ MRC complex I, II, IV; ↑ oxygen consumption. CC + NF-κB vs. CC: ↑ muscle mass; ↑ citrate synthase; ↑ MRC complex I, II, IV; ↑ oxygen consumption. CC + MAPK vs. CC: ↑ muscle mass; ↔ citrate synthase; ↔ MRC complex I, II; ↑ MRC complex IV; ↑ oxygen consumption |
| Fields et al. (2019) | Examine neural involvement in cachexia-linked respiratory insufficiency | Control; C26 | Normoxia: ↔ $V_T$; ↑ breathing frequency; ↑ $\dot{V}_E$ | Hypoxia: ↓ $V_T$; ↔ breathing frequency; ↓ $\dot{V}_E$; ↓ inspiratory burst amplitude; ↔ phrenic nerve firing frequency. Hypercapnia: ↔ $V_T$; ↔ breathing frequency; ↔ $\dot{V}_E$; ↔ inspiratory burst amplitude; ↔ phrenic nerve firing frequency. Maximal Chemoreflex: ↔ $V_T$; ↔ breathing frequency; ↔ $\dot{V}_E$; ↔ inspiratory burst amplitude; ↔ phrenic nerve firing frequency |
| Murphy et al. (2012) | Characterise functional impairments in mild and severe cachexia to inform the suitability of the C26 model | Control; C26-mild; C26-severe | CC-mild vs. control — Diaphragm: ↔ specific force; ↔ twitch characteristics; ↓ force (fatigued) | CC-severe vs. control: ↓ specific force; ↔ twitch characteristics; ↓ force (fatigued). CC-severe vs. control: ↓ specific force; ↔ twitch characteristics; ↓ force across force–freq. relationship |
| Murphy et al. (2013) | Assess whether treatment with perindopril enhances whole body and skeletal muscle function in cancer cachexia | Control; C26-mild; C26-mild + treatment; C26-severe; C26-severe + treatment | CC-mild vs. control — Diaphragm: ↓ specific force; ↔ twitch characteristics; ↔ specific force-freq. relationship | CC-mild vs. perindopril: ↔ specific force; ↔ twitch characteristics; ↓ specific force-freq. relationship. CC-severe vs. control: ↓ specific force; ↔ twitch characteristics; ↓ force across force-freq. relationship. CC-severe vs. perindopril: ↔ specific force; ↔ twitch characteristics; ↓ force across force–freq. relationship |

*(Continued)*

**Table 1. (Continued)**

| Author (Year) | Aim | Design | Respiratory relevant findings | | | |
|---|---|---|---|---|---|---|
| | | | Primary comparison | Alternative comparisons | | |
| Nosacka et al. (2020) | Assess pathophysiological differences between limb and diaphragm muscle in cancer cachexia | Control<br>PDAC-PDX | Diaphragm<br>↓ fibre CSA<br>↔ extracellular space<br>↑ irregular shaped fibres<br>↑ no. mononuclear cells<br>↑ no. necrotic fibres | Tibialis anterior<br>↓ fibre CSA<br>↔ extracellular space<br>↔ fibre shape<br>↔ no. mononuclear cells<br>↔ no. necrotic fibres | Transcriptome<br>No. genes upregulatedTA vs. DIA = 30 overlapNo. genes downregulated TA vs. DIA = 39 overlap | |
| Rosa-Caldwell et al. (2020) | Investigate signalling related to mitochondrial function, ROS production and protein synthesis during cancer cachexia development | Control – week 0<br>LLC – week 1<br>LLC – week 2<br>LLC – week 3<br>LLC – week 4 | Week 1 vs. week 0<br>↔ mitochondrial RCR<br>↔ mitochondrial content<br>↔ PGC-1α<br>↔ ROS production<br>↔ SOD1, SOD2 or SOD3<br>↔ FSR | Week 2 vs. week 0<br>↓ mitochondrial RCR<br>↔ mitochondrial content<br>↔ PGC-1α<br>↑ ROS production<br>↔ SOD1, SOD2 or SOD3<br>↔ FSR | Week 3 vs. week 0<br>↔ mitochondrial RCR<br>↔ mitochondrial content<br>↔ PGC-1α<br>↔ ROS production<br>↔ SOD1, SOD2 or SOD3<br>↔ FSR | Week 4 vs. week 0<br>↓ mitochondrial RCR<br>↔ mitochondrial content<br>↔ PGC-1α<br>↔ ROS production<br>↔ SOD1, SOD2 or SOD3<br>↔ FSR |
| Salazar-Degracia et al. (2018) | Assess the effects of treatment with $\beta_2$ agonist formoterol on atrophy signalling pathways and muscle metabolism of limb and respiratory muscle in cancer cachexia | Control<br>Control + formoterol<br>CC<br>CC + formoterol | CC vs. control<br>Diaphragm<br>↓ muscle mass<br>↓ muscle fibre CSA<br>↑ muscle structure abnormalities<br>↑ NF-κB activity<br>↑ FoxO activity<br>↑ myostatin levels<br>↓ mTOR activity levels<br>↓ PGC-1α | CC + formoterol vs. control<br>↓ muscle mass<br>↔ muscle fibre CSA<br>↓ muscle structure abnormalities<br>↓ NF-κB activity levels<br>↑ FoxO activity<br>↓ myostatin levels<br>↓ mTOR activity levels<br>↓ PGC-1α | CC + formoterol vs. CC<br>↔ muscle mass<br>↑ muscle fibre CSA<br>↓ muscle structure abnormalities<br>↓ NF-κB activity levels<br>↔ FoxO activity<br>↓ myostatin levels<br>↔ mTOR activity levels<br>↔ PGC-1α | |
| Smith et al. (2016) | Determine whether the JAK1/3 signalling pathway contributes to cancer cachexia-mediated diaphragm muscle weakness | Control<br>Control + JAK inhibitor<br>C26<br>C26 + JAK inhibitor | CC vs. control<br>Diaphragm<br>↓ specific force | CC + JAK vs. control<br>↔ specific force | | |
| Smuder et al. (2020) | Assess the impact of pharmacological treatment of mitochondrial dysfunction and ROS production in cancer cachexia | Control + saline<br>Control + SS-31<br>C26 + saline<br>C26 + SS-31 | CC vs. control<br>Diaphragm<br>↓ specific force<br>↓ fibre CSA<br>↓ normoxic $V_T$<br>↓ normoxic $V_E$<br>↓ hypoxic $V_T$<br>↓ hypoxic $\dot{V}_E$<br>↑ ROS emission<br>↓ mitochondrial RCR | CC + SS-31 vs. control<br>↔ specific force<br>↔ fibre CSA<br>↔ normoxic $V_T$<br>↔ normoxic $V_E$<br>↔ hypoxic $V_T$<br>↔ hypoxic $\dot{V}_E$<br>↔ ROS emission<br>↔ mitochondrial RCR | | |

↑, significant increase; ↓, significant decrease; ↔, no significant difference; CC, cancer cachexia; CSA, cross-sectional area; DIA, diaphragm; FEV1, forced expiratory volume in 1 s; FSR, fractional synthesis rate; LC, lung cancer; LLC, Lewis lung carcinoma; MRC, mitochondrial respiratory chain; NAC, *N*-acetyl cysteine; PDA, pancreatic ductal adenocarcinoma; PDAC-PDX, pancreatic ductal adenocarcinoma patient derived xenograft; RCR, respiratory control ratio; ROS, reactive oxygen species; SOD, superoxide dismutase; TA, tibialis anterior; VC, vital capacity; $V_T$, tidal volume; $\dot{V}_E$, minute ventilation.

concept reporting no atrophy of the diaphragm in mouse models of cachexia (Murphy et al., 2011; Tessitore et al., 1993). However, a follow-up study observed slight reductions in maximum specific force and absolute force in diaphragm strips obtained from C26 mice that lost 22% tumour-free body mass (Murphy et al., 2012). This study represents the earliest evidence of diaphragm muscle atrophy and contractile dysfunction in a mouse model of cancer cachexia. Supportive of this negative impact of cachexia, Roberts, Ahn and colleagues (2013) presented evidence of reduced respiratory muscle mass and function in C26 mice. The severe cachectic condition of these animals was associated with reductions in diaphragm muscle cross-sectional area of 24%, 22% and 29% in type I, type IIa and type IIb/x fibres, respectively, illustrating uniform wasting across fibre types. Additionally, mRNA expression of the atrophy-related biomarkers MuRF1 and MAFbx was elevated 2.8- and 2.2-fold in the diaphragm, respectively. Beyond muscle wasting, dysfunction of the diaphragm was evident in reduced submaximal and maximal diaphragm specific force production. In further elucidating the cause of such dysfunction, calcium-stimulated contractile characteristics were assessed in permeabilised single fibres. Findings were indicative of sarcomeric dysfunction and reduced calcium sensitivity, and this was accompanied by a 27% reduction in myosin heavy chain protein levels. Collectively, these results illustrate not only muscle wasting, but intrinsic dysfunction of the diaphragm beyond the observed decrease in muscle cross-sectional area. More recent investigations expand our understanding of the nature of the respiratory dysfunction present in cancer cachexia. Examining the neural drive of the phrenic nerve in C26 mice whilst manipulating inspired air, Fields et al. (2019) reported reduced inspiratory burst amplitude in response to hypoxia, but not hypercapnia or maximal hypercapnic/hypoxic response. This is suggestive of selective impairment of the neural mechanisms responsible for the hypoxic respiratory response, most likely at the level of hypoxic sensing. Similar to Roberts, Ahn et al. (2013) and Roberts, Frye et al., 2013), these findings highlight the complexity of the respiratory dysfunction beyond simple atrophy of the associated musculature.

Studies have highlighted an intuitive commonality in the mechanisms responsible for limb muscle wasting in cachexia, and those implicated in the generation of respiratory muscle weakness, atrophy and dysfunction. As is evident in studies concerned with cachexia and limb muscle, elevations in inflammatory factors have also been identified as a contributing mechanism of muscle atrophy and contractile dysfunction in ventilatory muscles (Janssen et al., 2005; Li et al., 2000). Additionally, the NF-$\kappa$B pathway was reported to be requisite for the transition of systemic inflammation to muscle atrophy in COPD-associated cachexia (Mittal et al., 2010). Pharmacological inhibition of either the NF-$\kappa$B or the MAPK pathway led to a restoration of both limb and diaphragm muscle mass and force in a lung cancer mouse model (Chacon-Cabrera et al., 2014). Mitochondrial oxidative stress is also suggested to be implicated in the multifactorial mechanisms of diaphragm atrophy and dysfunction (Duan & Bai, 2020). For example, oxidative stress induced by overexpression of Smad3 was associated with protein degradation in a ventilation-induced diaphragm dysfunction rat model. Inhibition of Smad3 prevented the protein degradation and reduced diaphragm contractility (Tang et al., 2017). Furthermore, there appears to exist an interconnected signalling network involving STAT3 signalling, Smad3 and FoxOs which contributes to diaphragm muscle atrophy and associated dysfunction, either directly or via mitochondrial dysfunction and oxidative stress (Duan & Bai, 2020). Interestingly, impaired mitochondrial function has been shown to present prior to the development of cancer cachexia in LLC mice (Brown et al., 2017), suggesting potential involvement in the development of the condition. These factors and signalling pathways that are implicated in limb muscle atrophy and altered by inflammatory and pro-oxidative states are of potential interest as therapeutic targets with the aim of minimising respiratory muscle atrophy and maintaining or restoring respiratory function. However, available information in cachexia-specific circumstances is limited and there is a distinct need for definitive profiling of the respiratory musculature under the imposed stresses of cachexia.

Further to the issue of limited respiratory focused studies in the context of cachexia, Nosacka and colleagues (2020) provided preliminary evidence to be considered in the generalisation of results obtained in the examination of limb muscle atrophy, which forms the focus of the majority of cachexia research. Transcriptome analysis suggested differential gene expression in tibialis anterior muscle compared to diaphragm muscle in response to the cancer-induced cachectic state. Additionally, while no apparent evidence of morphological damage was observed in limb muscle, consistent with past literature (Langen et al., 2012), histological analysis of diaphragm cryosections revealed disrupted muscle architecture, including increased extracellular space, irregularity of muscle fibre shape and necrotic fibres. Therefore, although commonalities are evident such as inflammation and oxidative stress, the manifestation of wasting and the response of the musculature to the stresses imposed may differ to a degree between limb and respiratory muscles. Such findings further emphasise the need for additional research aiming to characterise the profile of diaphragmatic muscle atrophy and dysfunction in cachexia.

## Benefits of exercise

The potential utility of exercise in reducing the severity of cachexia is promising. Exercise may provide both anti-inflammatory (Petersen & Pedersen, 2005) and antioxidative stimuli (Ji, 1999). Additionally, exercise type may impart a specific influence. For example, resistance exercise provides a potent hypertrophic stimulus in elevating levels of protein synthesis (Chesley et al., 1992) via activation of the mTOR pathway (Song et al., 2017). Alternatively, endurance exercise provides a potent metabolic stimulus by increasing mitochondrial biogenesis (Holloszy & Booth, 1976) via PGC-1$\alpha$ signalling (Pilegaard et al., 2006) (Fig. 3). The influence of such mechanistic stimuli can present in increased skeletal muscle strength and function, exercise capacity, and decreased fatigue, ultimately resulting in improved patient quality of life and handling of the condition (Knols et al., 2005; Schneider et al., 2007). Theoretically, these effects would act antagonistically to the fundamental mechanistic basis of the condition. Whereas cachexia is associated with an increase in protein degradation and concurrent decrease in protein synthesis, exercise may work to combat the adverse effects of the condition, whilst simultaneously increasing lean tissue mass. Contradictory to this, a recent study demonstrated that voluntary exercise was ineffective in improving the muscle phenotype of cachectic C26 mice (Hiroux et al., 2021). Additionally, it has been hypothesised that

increased energy demand due to tumour burden elevates the risk of over-reaching (Allan et al., 2022), an acute instance of insufficient recovery following exercise which if not recognised and accommodated for can progress to overtraining syndrome (Urhausen & Kindermann, 2002). In this way inappropriate implementation of exercise may exacerbate the cachectic state. These findings illustrate the need for careful implementation of a structured intervention defined by appropriate parameters should the benefits of exercise in cancer cachexia be optimally realised. Further research in this area is needed to elucidate such parameters and these investigations should be conducted in consideration of the knowledge transfer to clinical populations. However, prior to assessment of the specifics of intervention, additional foundational investigations into the mechanisms of effect of exercise in cachexia are needed.

**Tumour burden.** The influence of tumour burden on the cachectic state is considerable (De Lerma Barbaro et al., 2015). Reasonable preclinical evidence suggests exercise training may inhibit the progression of tumour growth and subsequently mitigate the impact of tumour burden on muscle tissue. For example, aerobic exercise training slowed tumour growth and attenuated cancer-induced muscle wasting in C26 mice (Khamoui et al., 2016). Similar observations have also been reported in Walker 256 breast tumour-bearing rats (Lira et al., 2008) and LLC tumour-bearing mice (Penna et al., 2011). The study of Pedersen et al. (2016) emphasised this beneficial effect of exercise, reporting a 60% reduction in tumour incidence and growth across multiple cancer models following voluntary wheel running. This effect was attributed to the acute immune response following exercise. Additionally, 16 days of high-intensity interval training has been shown to reduce tumour mass by 52% in LLC mice (Alves et al., 2018). This was accompanied by increases in mRNA levels of genes associated with inflammation and immunomodulation. These findings are particularly impressive considering the aggressive tumour progression and atrophy development of such cancer cachexia models. In addressing a considerable limitation of these studies, further research exploring the degree to which effects of exercise are maintained in orthotopic models is required. Such evidence would be informative as to the benefit of exercise in combatting tumour burden alongside the adverse impact of the associated tumour micro-environment.

**Hypertrophy/protein synthesis.** An intuitive potential benefit of an exercise intervention in combatting cancer cachexia is the stimulation of muscle hypertrophy, whereby activation of the associated pathways would act to create a shift in favour of protein synthesis and to

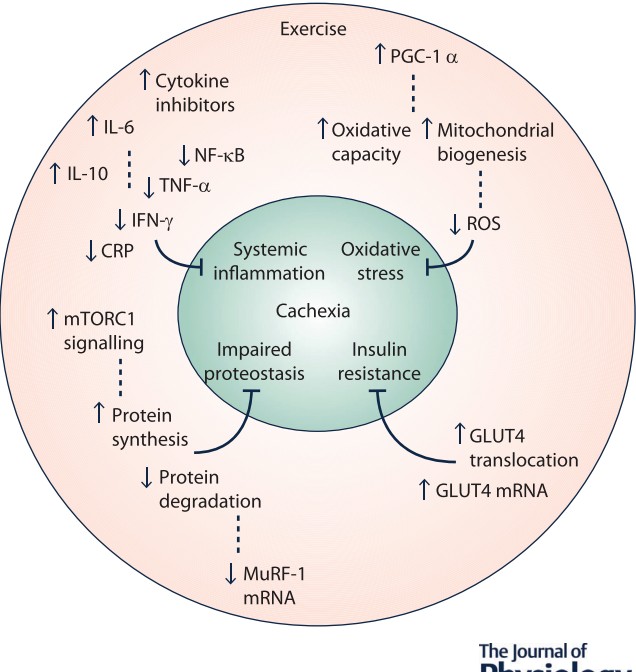

**Figure 3. The potential benefits of exercise in cachexia as reported in the literature**

disrupt and attenuate atrophic signalling. In this manner, regardless as to whether or not the stimulus is sufficient to produce considerable muscle growth, the mechanism of hypertrophic adaptation may reduce the severity of the cachectic condition by acting antagonistically to the mechanisms responsible for muscle atrophy and protein degradation. Resistance exercise training (RET) provides a potent stimulus toward such adaptation. Recent studies focusing on eccentric exercise training have provided evidence of its capability to activate mediating pathways of muscle growth, in addition to inhibiting mediators of muscle atrophy in preclinical cancer cachexia models. A maximal eccentric exercise intervention of 4 weeks resulted in elevated phosphorylation of p70S6K and rpS6, implying increased mTORC1 signalling in C26 mice. This was accompanied by a reduction in MuRF-1 mRNA and increased gastrocnemius muscle mass (Tatebayashi et al., 2018). An intervention consisting of eight sessions of repeated eccentric contractions in male APC$^{min/+}$ mice reflected similar results, with reduced muscle wasting, increased mTORC1 signalling and improved oxidative metabolism (Hardee et al., 2020). These results support the capability of exercise to reverse the imbalance of increased protein degradation and decreased protein synthesis as evident in the cachectic condition, helping to establish a state of proteostasis more compatible with the maintenance or recovery of muscle mass.

It is well established that eccentric contractions consume less oxygen and energy for a given muscle force than concentric contractions (Abbott et al., 1952). As a result eccentric RET is less energy demanding than both concentric RET (Lastayo et al., 1999) and endurance training (Bloomer, 2005), and hence, may be of particular benefit to those suffering from cachexia. Interest in the potential for imposing high loads on muscle tissue whilst reducing energy requirements led to the development of the Resistance Exercise via Eccentric Work (RENEW) modality, which advocates for moderate load eccentric exercise (LaStayo et al., 2014). Based on previous research, general protocol design would involve a frequency of 3 times per week, at an initial intensity of 50–75 W, for initial durations of 5–10 min. Progressive ramping of loads can then be introduced over 2–3 weeks without undue muscle soreness (LaStayo et al., 2017). Results support the use of this approach in producing similar gains in muscle strength and volume as traditional training, and it has been recommended in many clinical populations including cancer, sarcopenia and cachexia (Hoppeler, 2016; Julian et al., 2018; LaStayo et al., 2014). A range of techniques and equipment have been utilised to facilitate this form of eccentric training, including upper and lower limb ergometers (LaStayo et al., 2017), adapted treadmills to simulate downhill walking, or ergometers which mimic stair descending (Isner-Horobeti et al., 2013; Theodorou et al., 2013).

Notably, it is well established that eccentric contractions result in greater degrees of muscle damage in comparison to other contraction types (Fridén & Lieber, 1992). It is possible that excessive muscle damage may be detrimental to subsequent training efforts via decreased capability to produce force and increased levels of delayed-onset muscle soreness (Tee et al., 2007). Despite this, a reasonable body of evidence recommends eccentric training over concentric or isometric RET for muscle hypertrophy (Julian et al., 2018). However, the evidence in this regard is not conclusive, as a systematic review found similar changes in hypertrophy between eccentric and concentric training modalities (Franchi *et al.* 2017). Additionally, the implication of greater muscle damage in potentially enhancing the hypertrophic stimulus has been challenged, considering muscle damage is not essential to induce skeletal muscle hypertrophy and muscle becomes increasingly less susceptible to damage with repeated exercise (Nosacka et al., 2003). Ultimately, the optimal degree of exercise-induced muscle damage has not been determined, although it is hypothesised from past literature that a moderate degree should be sufficient in order to induce muscle hypertrophy (Schoenfeld, 2012). Should there be potential contraindications surrounding the implementation of eccentric training, submaximal contractions of progressive intensity, such as those employed in the RENEW modality, are recommended to minimise the likelihood of experiencing negative side effects (Hody et al., 2019).

**Anti-inflammatory effect.** Exercise is associated with the considerable production of anti-inflammatory cytokines (Ostrowski et al., 1999). First to appear in the circulation, IL-6 potentially increases 100-fold in response to exercise (Pedersen & Hoffman-Goetz, 2000; Steensberg et al., 2002), exhibiting a contrasting influence to that of its chronic presence in systemic inflammation. The anti-inflammatory influence of IL-6 is demonstrated in its antagonism of pro-inflammatory cytokine production, including TNF-$\alpha$ and IL-1 (Gould et al., 2013; Schindler et al., 1990). Additionally, IL-6 is implicated in the mediation of muscle growth (Serrano et al., 2008). As opposed to being correlated with reduced muscle cross-sectional area when chronically elevated, muscle growth induced by eccentric RET was positively correlated with IL-6 levels in APC$^{min/+}$ mice (Hardee et al., 2020). Exercise also results in increased levels of the anti-inflammatory cytokine IL-10, which inhibits the production of IL-1$\alpha$, IL-1$\beta$ and TNF-$\alpha$, and cytokine inhibitors, which act to block the signalling capability of pro-inflammatory factors (Helmark et al., 2010; Petersen & Pedersen, 2005). Considered to be the most practical, cost-effective and robust biomarker of cancer cachexia, C-reactive protein (CRP) is used to assess prognosis

and predict quality of life in cancer cachexia patients (Fearon et al., 2011). A meta-analysis of cancer-focused research found that combined resistance and endurance exercise training lead to reductions in CRP in cancer survivors (Khosravi et al., 2019). Additionally, 3 months of lower body RET was effective in reducing TNF-$\alpha$ levels in a frail elderly population (Greiwe et al., 2001). Similarly, RET reduced IFN-$\gamma$ levels after 12 weeks in elderly women (Roh et al., 2020), and after 8 weeks in prostate cancer patients (Papadopoulos et al., 2021). Finally, although not in a cachectic model, loaded ladder climbing, representing concentric RET, decreased NF-$\kappa$B expression in the inflammatory condition of Parkinson's disease in mice (Kim et al., 2021). However, it should be noted that many of these results were observed in non-cachectic cohorts and not in consideration of the subsequent impact on muscle atrophy.

**Anti-oxidative effect.** The mitochondrial oxidative pathway, which is of critical importance to cell metabolism and growth, is another physiological function impaired in cancer cachexia (VanderVeen et al., 2017), and is hypothesised to be implicated in the development of the condition. Exercise enhances the activity of antioxidant enzymes which play an important role in protecting against cell damage from ROS, with markers of damage being reduced following exercise (Khanna et al., 1998; Neuzil & Stocker 1993; Ohkuwa et al., 1997). Moderate intensity continuous exercise is known to stimulate metabolic alterations which increase antioxidant capacity in both humans and rodents (Gomez-Cabrera et al., 2008). Predicated on this knowledge, Ballaro et al. (2019) demonstrated the maintained antioxidative capabilities of exercise in the C26 mouse model. Here, moderate intensity treadmill running provided protection against muscle wasting and prevented loss of muscle function in association with reduced levels of ROS and improved oxidative capacity. Additionally, exercise may indirectly decrease ROS production by attenuating systemic inflammation (Bowen et al., 2015), as inflammatory factors such as TNF-$\alpha$ and IL-6 can promote a pro-oxidative state via the JAK–STAT3 signalling pathway (Bonetto et al., 2012; Powrozek et al., 2018). Furthermore, the activity of the NF-$\kappa$B and STAT3 pathways have been linked to mitochondrial dysfunction in muscle and thus are relevant therapeutic targets (VanderVeen et al., 2017).

Endurance training was recommended over RET for the reduction of muscle wasting by Aquila et al. (2020) in the context of oxidative stress in cancer cachexia. Indeed, endurance training is known to improve the oxidative capacity of muscle in healthy individuals by enhancing mitochondrial biogenesis and increasing glycogen content (Burgomaster et al., 2007, 2008). Considered in the context of cachexia, Mavropalias and colleagues (2022) suggested that the benefits of endurance exercise may be intensity dependent, with shorter duration, higher intensity training providing the most promise. Conversely, low-intensity continuous training is hypothesised to be a less than optimal approach due to its higher energy demands (Bloomer, 2005), potentially exacerbating the cachectic condition. In support of this concept, 8 weeks of high-intensity endurance training proved to be more effective in reducing NF-$\kappa$B levels than low-intensity endurance training in rats (Leite et al., 2021). However, a lack of primary studies assessing the effects of exercise training on cachexia limits our understanding of the impact that altering such exercise parameters can have on the condition. In contradiction of findings which advocate endurance training over RET for the reduction of oxidative stress (Aquila et al., 2020), STAT3 activation was reduced in the muscle of tumour-bearing cachectic mice following eight eccentric RET sessions across 2 weeks (Hardee et al., 2020) in addition to increasing muscle oxidative capacity and mitochondrial content (Hardee et al., 2016, 2020). Furthermore, basal protein synthesis was correlated with mitochondrial content (Hardee et al., 2020), suggesting a restorative effect on protein synthesis in association with maintenance of mitochondrial content and function.

Attempts to alleviate oxidative stress have also focused on the elevation of the transcription coactivator PGC-1$\alpha$, which primarily acts to increase mitochondrial content. However, findings in relation to the inhibition of muscle atrophy are mixed. For example, a large decrease in PGC-1$\alpha$ mRNA was observed in a mouse model of cancer cachexia, and transgenic overexpression of PGC-1$\alpha$ exhibited resistance to atrophy through the suppression of FoxO3 action (Sandri et al., 2006). Additionally, the same research group reported that the overexpression of an isoform, PGC-1$\alpha$4, facilitated the prevention of cancer cachexia in mice through the activation of IGF-1 and the repression of myostatin. However, employment of a similar transgenic model failed to illustrate a protective effect of PGC-1$\alpha$ overexpression in LLC mice (Wang et al., 2012), suggesting that targeting of increased mitochondrial biogenesis may be insufficient in the treatment of cancer cachexia.

Noting the distinct scarcity of research specific to a cachexia context, evidence from healthy and alternative disease populations suggest that RET and high-intensity endurance training are of potential benefit in combatting the adverse effects of cancer cachexia on mitochondrial function and the associated state of oxidative stress.

**Insulin resistance.** Peripheral insulin resistance is common in cancer cachexia due to decreased expression of glucose transporter type 4 (GLUT4) and muscle

glucose uptake (Puppa et al., 2014). This altered response to insulin likely contributes in some degree to reductions in protein synthesis (Baracos, 2000). TNF-$\alpha$ appears to bear a mediating influence in the development of insulin resistance in cancer (Noguchi et al., 1998), and so the anti-inflammatory effect of exercise may be of some benefit. However, likely to be a much more potent stimulus is exercise's ability through muscle contraction to increase expression and translocation of GLUT4 to the muscle membrane, facilitating uptake of glucose from the circulation independent of insulin (Flores-Opazo et al., 2020; Ren et al., 1994; Richter & Hargreaves, 2013). In cancer cachectic mice with reduced basal levels of GLUT4, a single 30-min session of electrically stimulated concentric contractions facilitated the increase of GLUT4 mRNA levels by 4.7-fold (Puppa et al., 2014). This offers evidence that this effect of exercise is maintained in the cachectic state. Insulin sensitivity has been increased in elderly men following a single session of eccentric RET per-week for 12 weeks, whereas concentric RET did not produce equivalent results (Chen et al., 2017). Further research conducted in alternative populations, including diabetic and healthy adults, implies that exercise reliant on high muscle glucose metabolism may be more efficacious in elevating GLUT4 expression, supporting the implementation of both high-intensity interval (Burgomaster et al., 2007) and low-intensity continuous (Bowman et al., 2021) endurance exercise, in addition to concentric RET (Dehghan et al., 2016). The obvious issue in implementing an intervention aimed at maximising glucose metabolism is a higher energy yield is undesirable in the cachectic individual. Nonetheless, muscle contraction-induced mobilisation of GLUT4 would likely be a beneficial concomitant effect of an exercise intervention aimed at combatting cachexia. Additionally, exercise is known to stimulate angiogenesis (Andersen & Henriksson, 1977), largely via the upregulation of vascular endothelial growth factor (Breen et al., 1996). This is typically observed in response to endurance exercise (Hudlicka et al., 1992). Both chemical and mechanical mechanisms are potentially implicated, including increases in stretch and compression, shear stress and transmural pressure of the vessels (Kissane & Egginton, 2019), in addition to elevations in hypoxia-inducible factor in response to low partial pressure of oxygen in the blood (Shweiki et al., 1992). Significant increases in capillary density can be observed following only 4 weeks of cycle ergometer training at ∼ 60% maximal oxygen uptake (Hoier et al., 2012), and this timeline is reflected in rodents following voluntary wheel running (Waters et al., 2004). Such an adaptation may further enhance insulin sensitivity through improved blood–tissue exchange properties. A greater capillary network increases the surface area for

diffusion, shortens the average diffusion pathway length, and increases the length of time for diffusive exchange between the blood and the muscle (Bloor, 2005). Of note, enhancement of blood exchange properties is dependent on the relative ratio between capillary surface area and muscle fibre surface area (Hepple et al., 2000). In this regard RET has been suggested to be less beneficial or perhaps even detrimental towards this adaptation as hypertrophy of muscle fibres can decrease this ratio (Tesch et al., 1984). However, considering the population in question, any considerable skeletal muscle hypertrophy would far outweigh the benefits which enhanced blood–tissue glucose exchange may have on insulin resistance.

From this brief overview of the literature, we conclude that exercise training has the potential to exert many beneficial effects which may culminate to counteract the adverse effects of cachexia. However, many of these effects are yet to be comprehensively investigated in cachexia specifically, with much of the above information originating from alternative populations. Therefore, although as much information as possible should be gleaned from research into conditions which elicit similar wasting profiles, it is important to acknowledge that all of the effects of exercise may not hold true in cachexia. As a result, further attempts to contribute to the knowledge base surrounding the mechanistic alterations of the cachectic state resulting from exercise would be of considerable value in advancing this area. In order to ensure valid and practically meaningful results are obtained from such investigations an appropriate model of cancer cachexia must be utilised.

## Preclinical models of cancer cachexia

There are a number of preclinical methodological options available designed to mimic the human cancer cachectic condition in mice. These include subcutaneous, orthotopic, chemically induced and genetically engineered variations. The classically employed models in the study of cancer cachexia are the subcutaneous C26 and LLC models. An insightful systematic review highlighted the popularity of these models. Compared to orthotopic and genetically engineered mouse models (GEMM), subcutaneous models were found to be the most abundant in the early research, with the C26 variation continuing to be predominant in 2017 (Tomasin et al., 2019). Results from the extensive use of subcutaneous models have facilitated the advancement of our understanding of the mechanisms responsible for muscle atrophy in the condition. However, the clinical translatability of these models has been questioned (Johns et al., 2013). It should be acknowledged that no single model fully recapitulates

cancer cachexia in humans (Mueller et al., 2016). However, continued model development with the aim of addressing certain limitations has led to the production of viable alternatives, particularly with regard to the representation of pancreatic cancer cachexia (Delitto et al., 2017; Henderson et al., 2018). Consideration concerning which model to employ should primarily focus on the nature of the resultant cachexia phenotype to ensure that this is in line with the focal point of the research to be conducted and optimises the translation of findings to the clinical setting. Additionally, the associated benefits and drawbacks of each model should also be considered.

Subcutaneous models are an ectopic model, involving the injection of cultured tumour-derived cells into the flank of the animal which have been reported to be representative of the cancer cachectic condition (Aulino et al., 2010). Supporting the use of the C26 model in juvenile mice, comparable tumour development and proteolytic signalling was observed between 8-week-old and 12-month-old animals (Talbert et al., 2014). Notably, 12-month-old mice were used in this study as this age is before the onset of sarcopenia (Sayer et al., 2013), accounting for this potential confounding variable. Slight differences were reported with young mice experiencing ∼10% greater limb muscle mass loss, which was attributed to a greater relative tumour burden as evident from the tumour mass to body mass ratio (Talbert et al., 2014). Highlighted as an issue of the C26 and LLC models is the typically large resultant tumours, which often represent >10% of whole-body mass (Talbert et al., 2017; Zhang et al., 2017). This reflects one of the primary criticisms of these models, in that they may represent an inhibition of growth in young animals, as opposed to the active atrophy of established muscle mass seen in cachexia (Talbert et al., 2019). This is likely due to not only the use of younger animals, but also the aggressive nature of the models whereby the induction and development of cachexia occurs over a time scale of only a few weeks (Roberts, Ahn et al., 2013; Talbert et al., 2014). This rapid onset of wasting may be a reflection of the reliance of subcutaneous models on IL-6 and the induction of an inflammatory phenotype (Bonetto et al., 2011). Although systematic inflammation is associated with and contributes to cachexia progression, the levels observed in subcutaneous models far exceed those of cancer patients (Lerner et al., 2015; Strassmann et al., 1992; Talbert et al., 2018). Furthermore, this narrow window from induction to death shortens the experimental period, limiting the analysis of potential treatments. Another acknowledged limitation of the subcutaneous options, as ectopic models, is the development of the tumour in a non-physiological growth environment (Bonetto et al., 2016), and hence, the absence of a representative tumour microenvironment,

which constitutes a proportion of the disparity between these preclinical models and human cancer patients. This represents one of the primary advantages for which one might consider employing an orthotopic cancer cachexia model.

Orthotopic models are frequently employed and recommended for the study of pancreatic cancer cachexia (Qiu & Su, 2013). Such models involve implantation of human or mouse cancer cell lines or tumour tissue into a syngeneic site. This facilitates the establishment of an organ-specific tumour microenvironment, allowing the development of site-specific pathology (Go et al., 2017). In this way, orthotopic models are suggested to be more clinically relevant than subcutaneous models (Delitto et al., 2017; Gengenbacher et al., 2017). Variations of orthotopically induced pancreatic cancer cachexia models have reported results supportive of their ability to reflect numerous aspects of the condition (Henderson et al., 2018), including decreased body mass and limb muscle CSA, upregulated cachexia biomarkers, including FoxO proteins and E3 ubiquitin ligases, and elevated levels of pro-inflammatory cytokines (Greco et al., 2015; Jones-Bolin & Ruggeri, 2007; Shukla et al., 2015). However, orthotopic models are also associated with certain limitations, primarily the required use of immuno-deficient animals to prevent rejection of the transplanted tissue (Henderson et al., 2018), which likely influences the pathophysiological response of the animal to the condition.

Recently, Talbert and colleagues (2019) designed a novel GEMM of pancreatic ductal adenocarcinoma (PDA), with the intention of accurately recapitulating the human cancer cachectic phenotype, in consideration of the limitations of the C26 and LLC models. The resultant model, named KPP, is reported to express a number of improvements in this regard. Modification of the Kras–p53–Cre model (Parajuli et al., 2018) allows induction of PDA via tamoxifen administration. Characterisation of the model at 60, 75 and 90 days post-induction illustrated a progressive nature of muscle wasting, with decreases in body weight matching observed decreases in skeletal muscle, heart and adipose mass relative to controls. Interestingly, the tibia lengths of KPP mice were comparable to control animals, suggesting that the observed muscle atrophy was not due to a stunting of development, which is a hypothesised limitation of subcutaneous models. Notably, a comparable phenotype was observed in 1-year-old animals, whereby tamoxifen administration induced progressive PDA development over a median post-injection survival period of 158 days. Importantly, un-injected KPP mice showed no apparent pancreas pathology. In comparison to the rapid onset of muscle wasting induced in many other preclinical models, the progressive development of the cachectic

condition in the KPP model more accurately represents the nature of muscle atrophy observed in cancer patients (Johns et al., 2013). In this way the KPP model could potentially facilitate research aimed at exploring the initial mechanistic alterations that occur in the developmental stages of cancer cachexia, an area of study recommended for the improvement of diagnosis and prevention of the condition (Rosa-Caldwell et al., 2020). Additionally, these features would allow extensive assessment of therapeutic interventions. For example, the potential benefit of chronic exercise training or training performed prior to the onset of wasting could be explored. In consideration of the variation in cachexia severity dependent on tumour type and site (Dewys et al., 1980), Talbert and colleagues acknowledged the potential for features of the KPP model to be selective to PDA. Nevertheless, the characteristics of this model present it as a promising tool in the future study of cachexia, particularly in the context of muscle atrophy associated with pancreatic cancer, which considering the prevalence reported in this population (Fearon et al., 2011) is an area of utmost importance.

## Conclusion

Cachexia is a severe condition associated with the progression of chronic illness, bearing a notable impact in cancer considering its prevalence in this population. Use of the classification of cachexia as a prognostic marker in cancer patients highlights the severe adverse effect imposed during the development of this condition. Elevations in the morbidity and mortality rates of cancer patients suffering from cachexia has been linked to respiratory failure subsequent to atrophy and dysfunction of the ventilatory muscles. Despite this, there is a distinct scarcity of research investigating the structural and functional condition of the respiratory musculature in response to the stresses of cachexia, with the vast majority of studies exclusively focusing on limb muscle. Considering some of the findings of the small minority that have addressed this issue with a respiratory focus, this area of research warrants much further attention. Although a considerable number of reports have been published investigating mechanisms of action of cachexia in limb muscles, research into the presentation of the condition in the respiratory musculature would be valuable due to the importance of this system in disease outcomes and potential differences in response between limb and diaphragm muscle. Therefore, profiling of cancer cachexia in the respiratory musculature would be a valuable pursuit. Furthermore, such results would aid the identification of therapeutic targets, facilitating subsequent investigation into efficacious intervention design for the treatment of the condition. It is accepted that

such an intervention would be multi-factorial, combining elements of pharmacology, nutrition and exercise. Further research is required into each of these components. However, exercise has received comparatively little attention. Considering the presentation of cachexia and the known effects of exercise in healthy individuals in addition to disease populations, it is probable that further research into its bearing on the cachectic state will yield valuable results with potential application to disease management. In time, such research will facilitate exploration into the influence of varying exercise parameters on patient response, allowing optimisation of intervention protocols. Additionally, further understanding of the response of the cachectic individual to exercise in isolation would benefit future studies aiming to design, implement and assess multifactorial therapeutic approaches.

## Appendix A: Inclusion and exclusion criteria

### Inclusion

Studies were considered for inclusion provided they fulfilled the following criteria.

(i) Participants/subjects included either patients suffering from cancer cachexia or preclinical models of cancer cachexia.
(ii) There was assessment of the impact of cancer cachexia on the respiratory system using primary measures, including, but not limited to, measures of respiratory muscle wasting and/or function.
(iii) There was inclusion of an appropriate control group for comparison.
(iv) The study was published in English in a peer-reviewed journal.

### Exclusion

Studies were excluded if they fulfilled the following criteria.

(i) They including other wasting conditions, where isolation of cachexia data is not possible.
(ii) They presented unpublished data or data only available in abstract form.
(iii) They were non-primary studies such as reviews and meta-analyses.

## Appendix B: Search strategy table

B1

**Table B1. MeSH headings and text words which were combined to generate search strategies**

| Population | | | Intervention | Comparison | Outcome |
|---|---|---|---|---|---|
| *Medical subject headings (MeSH)* | | | | | |
| Cachexia | Mice | Human | Exercise | | Diaphragm |
| | | Adult | | | Respiratory muscles |
| | | Clinical study | | | Respiratory system |
| | | Clinical trial | | | |
| *Text words* | | | | | |
| Cancer | Mouse model | | Exercise | | Respiratory |
| Cancer cachexia | Murine | | Exercise intervention | | Respiratory function |
| Colon adenocarcinoma 26 | | | Training | | Respiratory control |
| Lewis lung carcinoma | | | Aerobic training | | Diaphragm function |
| | | | Aerobic exercise | | Diaphragm control |
| Pancreatic ductal adenocarcinoma | | | Resistance training | | Muscle atrophy |
| LLC | | | Resistance exercise | | Muscle wasting |
| C26 | | | | | |
| PDA | | | | | |

## Appendix C: PubMed search strategy

(1) ((Cachexia[mh]) OR (cancer[tiab] AND cachexia[tiab]) OR cachexia[tiab])

(2) (Mice[mh] OR 'mouse model'[tiab] OR murine[tiab] OR 'Colon Adenocarcinoma 26'[tiab] OR C26[tiab] OR 'Lewis Lung Carcinoma'[tiab] OR LLC[tiab] OR 'Pancreatic Ductal Adenocarcinoma'[tiab] OR PDA[tiab])

(3) (Human[mh] OR Adult[mh] OR 'Clinical Study'[mh] OR 'Clinical Trial'[mh] OR 'clinical trial'[tiab])

(4) (Diaphragm[mh] OR 'Respiratory muscles'[mh] OR 'Respiratory system'[mh] OR Respiratory[tiab] OR 'Respiratory function'[tiab] OR 'Respiratory control'[tiab] OR 'Diaphragm control'[tiab] OR 'Diaphragm function'[tiab] OR 'respiratory muscle atrophy'[tiab] OR 'respiratory muscle wasting'[tiab])

(5) (1) AND (2)

(6) (1) AND (3)

(7) (1) AND (2) AND (4)

(8) (1) AND (3) AND (4)

(9) (7) OR (8)

## Appendix D: PRISMA flow diagram

Figure D1.

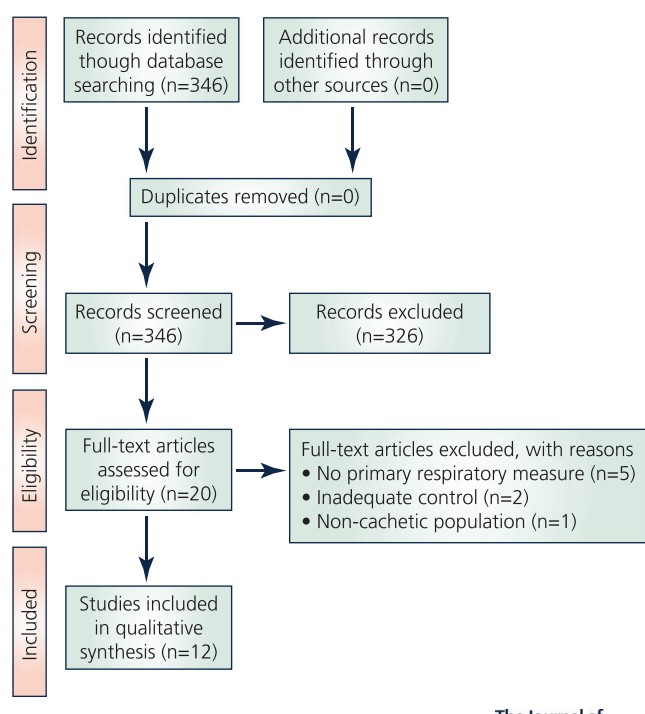

**Figure D1. PRISMA flow diagram illustrating the number of papers excluded at each stage of the screening process (Moher et al., 2009)**

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

## Additional information

### Competing interests

None.

### Author contributions

B.M. and K.O'H. were responsible for the concept of the article. B.M. wrote the initial draft of the text. All authors contributed to the critical evaluation and revision of the text for important intellectual content. All authors have read and approved the final version of this manuscript and agree to be accountable for all aspects of the work in ensuring that questions related to the accuracy or integrity of any part of the work are appropriately investigated and resolved. All persons designated as authors qualify for authorship, and all those who qualify for authorship are listed.

### Funding

None.

### Acknowledgements

### Keywords

cachexia, cancer, exercise, respiratory system

## Supporting information

Additional supporting information can be found online in the Supporting Information section at the end of the HTML view of the article. Supporting information files available:

**Peer Review History**

